# Stakeholders’ Understandings of Human Papillomavirus (HPV) Vaccination in Sub-Saharan Africa: A Rapid Qualitative Systematic Review

**DOI:** 10.3390/vaccines9050496

**Published:** 2021-05-12

**Authors:** Caroline Deignan, Alison Swartz, Sara Cooper, Christopher J. Colvin

**Affiliations:** 1The Division of Social and Behavioural Sciences, School of Public Health and Family Medicine, University of Cape Town, Cape Town 7935, South Africa; alison.swartz@uct.ac.za (A.S.); cj.colvin@uct.ac.za (C.J.C.); 2South African Medical Research Council, Cape Town 7501, South Africa; Sara.cooper@mrc.ac.za

**Keywords:** Human Papillomavirus (HPV), vaccination, Sub-Saharan Africa (SSA), stakeholder understandings, rapid qualitative systematic review, thematic analysis

## Abstract

Cervical cancer rates in Sub-Saharan Africa (SSA) are amongst the highest worldwide. All three of the Human Papillomavirus (HPV) vaccines (9-valent, quadrivalent and bivalent HPV vaccine) provide primary protection against the most common cancer-causing strains of HPV (types 16 and 18) that are known to cause 70% of cervical cancers. Over the last five years, there has been an increase in Sub-Saharan African countries that have introduced the HPV vaccine. The majority of research has been conducted on supply-side barriers and facilitators to HPV vaccination uptake in SSA, yet little research has been conducted on demand-side or end-user perspectives of, and decisions around, HPV vaccination. In order to complement existing research, and inform current and future HPV vaccination implementation approaches, this qualitative systematic review explored Stakeholders’ understandings of HPV vaccination in SSA. This review searched the following databases: Embase (via Scopus), Scopus, MEDLINE (via PubMed), PubMed, EBSCOhost, Academic Search Premier, Africa-Wide Information, CINAHL, PsycARTICLES, PsycINFO, SocINDEX, Web of Science, and the Cochrane Controlled Register of Trials (CENTRAL) and found a total of 259 articles. Thirty-one studies were found eligible for inclusion and were analyzed thematically using Braun and Clarke’s methods for conducting a thematic analysis. The quality of included studies was assessed using the Critical Appraisal Skills Programme (CASP) checklist. Three major themes emerged from this analysis; knowledge of HPV vaccination and cervical cancer is intertwined with misinformation; fear has shaped contradictory perceptions about HPV vaccination and gender dynamics are relevant in how stakeholders understand HPV vaccination in SSA.

## 1. Introduction

Human papillomavirus (HPV) is a central causative agent of cervical cancer, the second most common cancer among women in Africa, where approximately 372.2 million women aged 15 years and older are at risk of developing cervical cancer [1]. There are more than 100 types of HPV, at least 14 of which are high-risk strains that can cause cancers of the vulva, vagina, penis, anus, oropharynx and most commonly the cervix [2,3,4]. Such strains are commonly sexually transmitted an are highly transmissible through skin-to-skin genital contact, anal, oral and vaginal sex, with estimations that two thirds of those who have had sexual contact with HPV-infected persons will become infected themselves [2,5], making it the most common viral infection of the reproductive tract and the most common sexually transmitted disease in the world [2,5,6].

The quadrivalent HPV vaccine was first approved by the U.S Food and Drug Administration (FDA) in 2006, offering primary protection against the most common cancer-causing strains of HPV (types 6, 11, 16, 18) followed by the bivalent vaccine approved in 2009 (protection against type 16 and 18) (both are which are primarily used in SSA), and finally the 9-valent vaccine approved in 2014 (providing protection against 6, 11, 16, 18, 31, 33, 45, 52, and 58). In SSA, late presentation to care is an established trend associated with health inequities, gender disparities, socioeconomic and cultural factors that have ultimately increased cervical cancer mortality rates [3,5]. The World Health Organization (WHO) estimates that at least one third of all HPV-related cancers in Africa could be prevented with comprehensive vaccination implementation [3].

As of June 2019, nine Sub-Saharan African countries have included HPV vaccination in their National Immunization Programs (NIPs): Botswana (2015), Lesotho (2012), Rwanda (2011), Sao Tome and Principe (2016), Senegal (2016), Seychelles (2014), South Africa (2014), Uganda (2012), and Mauritius (2016) [1,7,8,9]. Twenty-two SSA countries have HPV vaccine demonstration projects in place. These numbers are likely to increase considerably—over the last 5 years, the Global Alliance for Vaccines and Immunization (GAVI), along with the World Health Organization, PATH and UNICEF, has generated substantial momentum around HPV vaccination in SSA [10].

In order for any type of vaccination to be successful, high levels of uptake are required in order to promote herd immunity, a phenomenon where enough people in the population are vaccinated and the pathogen cannot reproduce [11]. Despite a steady rise in the number of African countries introducing HPV vaccination into National Immunization Programs and demonstration/pilot projects, coverage is still suboptimal [8]. According to GAVI, all five of the countries with the highest numbers of deaths from cervical cancer remain in SSA, highlighting the major public health concern of the continued prevalence and incidence of cervical cancer and the need for increased uptake of HPV vaccination in SSA [12].

HPV vaccination coverage in SSA is shaped by a range of factors [4]. The majority of research in the region has tended to focus on health system and supply-side barriers, including health system capabilities, inaccessibility to medical care, low cervical cancer screening levels, inadequate infrastructure, finances, and health worker training as significant systemic barriers to HPV vaccination success [8,13,14,15,16]. However, there is growing recognition that demand-side barriers may also be contributing to suboptimal HPV vaccination coverage in SSA [2,11,17]. Persistent skepticism around HPV vaccination has contributed to stakeholder uncertainty in vaccination behavior and actual decision making around the HPV vaccine. This has motivated further research into understanding why people get vaccinated and why they do not, especially given “the incredible impact and value that vaccination has had” and the continued lack of clarity around low levels of HPV vaccination uptake in countries where it has become available and affordable [18].

This review focuses on the demand-side barriers and facilitators of HPV vaccination by examining the perspectives and understandings of relevant stakeholders involved in decision-making linked to the uptake of HPV vaccination in SSA. This review explores the qualitative literature documenting the perspectives of adolescents, parents and caregivers, teachers and health care providers, and political, community and religious leaders in creating ‘demand’ for HPV vaccination. By exploring an ‘end-user’ or ‘demand-side’ point of view from these stakeholders, valuable insight into factors that promote or inhibit HPV vaccination efforts in SSA may become clearer. Utilizing empirical qualitative evidence from various demand-side stakeholders’ perspectives will more accurately reflect the spectrum of HPV vaccination behavior and decision-making in SSA.

## 2. Materials and Methods

### 2.1. Review Question, Aims and Objectives

This rapid systematic review explored how stakeholders understand the HPV vaccine and HPV vaccination in SSA. The key objectives included identifying, appraising and synthesizing the qualitative evidence of stakeholders understandings, experiences, and perceptions of the HPV vaccine and HPV vaccination in SSA in order to complement reviews on HPV vaccine effectiveness and contribute to understandings of the barriers and facilitators of successful implementation of HPV vaccination strategies in SSA [19].

### 2.2. Search Strategy

This review conducted systematic and comprehensive searches during March and April 2019 using the following relevant databases: Embase (via Scopus), Scopus, MEDLINE (via PubMed), PubMed, EBSCOhost, Academic Search Premier, Africa-Wide Information, CINAHL, PsycARTICLES, PsycINFO, SocINDEX, Web of Science, and the Cochrane Controlled Register of Trials (CENTRAL). The search terms related to HPV and the HPV vaccine, terms related to comprehension and understandings, terms related to SSA, and terms related to qualitative research (Appendix A). The reference lists of systematic reviews identified through the database searches were scanned in order to identify additional relevant papers.

### 2.3. Inclusion and Exclusion Criteria

This review sought studies on stakeholder ‘understandings’ of the HPV vaccine in SSA, broadly defined, including studies on understandings, experiences, perceptions, attitudes, beliefs, knowledge, comprehensions, feelings, and opinions of the HPV vaccine in SSA. The population of interest included relevant stakeholders who create a ‘demand’ for the vaccine including adolescents, parents and caregivers of adolescents, teachers, health care providers, and political, religious, and community leaders.

Eligible research articles took place in SSA and were published in English from 2006 (when the first HPV vaccine Gardasil was manufactured)-present. Eligible articles used a qualitative or mixed methods study design, where both data collection and analysis employed qualitative methods.

Studies outside of SSA were excluded. Studies examining HPV-related health outcomes, or vaccination more generally without a specific focus on the HPV vaccine, were also excluded, as were quantitative studies and systematic reviews. Grey literature and non-English studies were excluded.

### 2.4. Study Selection

Studies were selected through a 4-stage process where titles and abstracts were identified through hand and database searches and were then screened for eligibility by one author. To maintain rigor, 20% of eligible title and abstracts were screened by two independent reviewers and were reviewed for full text eligibility. A final list of eligible studies was collated and uploaded into Covidence software for quality assessment and data extraction. The search process is outlined in the PRISMA diagram (Figure 1) and the included studies were mapped (Figure 2).

### 2.5. Data Extraction

Qualitative data from each study were extracted into a Microsoft Excel sheet. Relevant contextual information from each study was also extracted, including first author name, year of study, year of publication, the country where the study took place, the study participants, the qualitative data collection and analysis methods, and the aim of the study (Appendix B).

### 2.6. Study Quality Assessment

All included studies were assessed by lead author (CD) using the Critical Appraisal Skills Programme (CASP) appraisal tool for quality assessment purposes. The CASP checklist includes the following criteria: aims of the research question, the qualitative methodology, the research design, the recruitment strategy, data collection, reflexivity, ethical issues being considered, data analysis and findings being of substantial value and evidence. Studies were not excluded based on the CASP appraisal tool assessment alone (Appendix C), but this assessment informed the analyses and overarching findings of the review.

### 2.7. Data Extraction and Analysis

A thematic analysis approach, as described by Braun and Clarke (2006), was used to synthesize the data. Thematic analysis develops and organizes findings into themes using inductive and ‘constant comparison’ methods and is an appropriate synthesis method for exploring questions about people’s perspectives and understandings [21,22]. The overall themes and the components of each individual theme (made up of sub-themes) were confirmed and named. This contributed to the greater narrative of stakeholder understandings, experiences and perceptions of the HPV vaccine in SSA. The analysis of this review identified potentially relevant themes, schools of thought and actions for consideration of HPV vaccination roll out based on the contexts in which the studies took place.

## 3. Results

After deduplication, abstract and title screening and full text screening, thirty-one eligible studies were included in the review.

### 3.1. Study Characteristics

As shown in Figure 2, the majority of included studies came from Eastern and Southern Africa, predominately from Uganda, Kenya, and South Africa. All of the studies utilized focus group discussions or interviews (or both) for data collection methods. Thematic, comparative analysis, narrative synthesis, grounded theory and content analysis were the four methods of data analysis across studies (Appendix B for more information about the studies).

### 3.2. Findings

Please see Summary of themes and findings in Table 1.

### 3.3. Theme 1: Knowledge and Misinformation

Stakeholders have knowledge but also misinformation in relation to understandings of cervical cancer risks and transmission that shape perceptions of HPV vaccination as a whole. This knowledge and misinformation has left many stakeholders uncertain about how to understand and make decisions linked to HPV vaccination.

### 3.4. Understandings and Perceptions of Cervical Cancer and Its Causes

There were generally low levels of technical knowledge about cervical cancer and HPV in SSA [14,23,24,25,26,27]. This was most evident in stakeholders not knowing that HPV is the causative agent of cervical cancer, followed by stakeholder confusion and uncertainty around which part of the female reproductive tract is affected by cervical cancer [14,15,25,26,28,29,30], further evidenced in the quote, “*I have never heard about cancer of the cervix. What is the difference between the cervix and the uterus? We thought it was the same thing*” from one female interviewed in Uganda [31]. In general, stakeholders seldom made distinctions between the cervix, uterus, and womb when discussing cervical cancer, most often referred to as ‘cancer of the womb’ in South Africa, Kenya, and Uganda [14,15,23,31,32,33]. Across studies, “cancer of the uterus” was the second most common reference to cervical cancer, highlighting explicit confusion between the cervix and the uterus [13,23,29,31,34].

How people speak about a health topic is often a good indication of what they understand about it [35]. When stakeholders were asked what they knew about cervical cancer, responses consistently included explanation or recognition of the disease through perceived symptoms (accurate or inaccurate), rather than the use of correct terminology to name the disease and its cause, emphasizing nuances in local language [31]. Further evidence of stakeholder confusion around cervical cancer and its causes were evident in some stakeholders explaining they feel they are “in the dark” about HPV and are anxious to know more about it [14,26].

The majority of stakeholders across included studies had limited technical knowledge about cervical cancer and HPV. Many were aware that HPV is sexually transmitted and cited common symptoms of cervical cancer including vaginal bleeding, foul odor, vaginal discharge, and pain during intercourse [26,36,37,38]. Few stakeholders, usually health care providers, clinicians, and educators, were aware that HPV can be a silent infection that affects men and women alike [25,36,39]. Stakeholders often insinuated that promiscuity and poor hygiene were the cause of cervical cancer [23,25,30,37]. As a Nigerian male religious leader explained, “*If a woman has sex anyhow with different men, it can cause cancer... If you sleep with 5 men that is 5 different diseases, 8 men means 8 different diseases…. So, if a woman sleeps around she can have cervical cancer*” [25]. Others attributed sexual activity to the accumulation of sexually transmitted infections, that when left untreated, cause a block in a woman’s reproductive tract that produces cancer from a “combination of several diseases” [23].

Stakeholders who personally knew someone with cervical cancer were less likely to link cervical cancer aetiology to promiscuity. Some expressed cognitive dissonance in knowing “conservative” women who were diagnosed with cervical cancer [23,25]. Additional skepticism around a male being an asymptomatic carrier of HPV added to confusion around the causes of cervical cancer and the aetiological link to HPV [23,25].

Other perceived or hypothesized causes that stakeholders attributed to cervical cancer were: a modern lifestyle of processed food and reliance on medication [23,25,31], genetics [23,25,40], witchcraft [37], vaginal insertion of drugs [26,37], progression of existing STDs [23], lack of cleanliness [33], having sex while menstruating [33], penile impurities [33], poor sexual habits [37] and frequent childbirth or frequent abortion [26].

### 3.5. Understandings of the Risks for HPV and Cervical Cancer

Cancer in general was widely feared and stigmatized by participants across studies [14,25,26,33,41]. When asked about perceptions of cervical cancer, stakeholders across studies described it as “painful”, “dangerous”, “horrible”, “incurable”, “deadly”, and a “death sentence” [29,30,33,38]. Words like “death”, “suffering”, “pain”, and “reduced quality of life” were commonly associated stakeholders talking about a cervical cancer diagnosis [14,25,26,41]. Such perceptions were often based on stakeholders’ experience of knowing someone with cervical cancer and having watched them suffer from the disease [14,23,25,29,30,38].

When discussing both cervical cancer diagnosis and HPV, stakeholders highlighted fear as a predominant emotion [14,25,26,41]. The fear of cervical cancer was so great that some women expressed reluctance to seek care, saying they would prefer “not to know” as they believed cervical cancer would inevitably lead to death [38,41,42,43] evident in a female participant in a focus group discussion in Uganda expressing: “*I fear going for a check-up since after getting the diagnosis of cervical cancer, I will then know that I am dying*” [38]. This was also supported in South Africa where women expressed that they would rather not “face” the possibility of a cervical cancer diagnosis [44].

In most studies, female stakeholders perceived a high risk of cervical cancer for both themselves and their female counterparts [30,31,32,38]. This risk perception was linked to high rates of cervical cancer mortality in stakeholder’s country context, thus reinforcing a clear desire from stakeholders (female and male) to avoid a cervical cancer diagnosis in oneself and loved ones [14,15,25,26,29,30].

The fear of cervical cancer, the pain associated with it, and the desire to avoid it were frequently mentioned across studies. This led stakeholders who understood the purpose of HPV vaccination, to conduct a risk assessment, weighing up the short-term pain (of the HPV vaccine injection) with long term pain (of potential cervical cancer diagnosis), often reaching the conclusion that “prevention is better than cure” [14,15,25,26,29,30] with a Ugandan female adolescent explaining “*I have had injections before and I know the pain lasts a short time but I feared that the pain from cancer can last forever*” [29].

### 3.6. Understandings of HPV Vaccination

Across studies there was a lack of awareness that an HPV vaccine exists [14,15,25,26,28,29,30]. Despite low levels of baseline technical knowledge about the HPV vaccine, there was notable willingness to learn more about HPV vaccination [14,24,29,45,46].

Preventing cancer was a large part of the rationale for getting vaccinated against HPV, sometimes referred to as the “cancer of the womb vaccination” in countries such as South Africa [14]. A Ugandan adolescent female explained: “*People say that cancer has no cure but the health worker told us that this cancer [HPV]… could be prevented, so I was very happy because it meant that if I got vaccinated I would not die of cancer*” [29].

Although there was clear interest in preventing cervical cancer, many stakeholders raised the point that without symptoms, they were unsure why HPV vaccination was necessary, thus highlighting a lack of knowledge around the notion that vaccines prevent the onset of disease. This sentiment was especially evident in one participant from a focus group in Kenya who asked: “*What are you vaccinating us against, yet we are not sick?*” [28] which reinforced symptomology as a driver in health seeking behavior [36,37,38]. Without tangible symptoms, stakeholder action towards vaccination uptake may be inhibited, illustrated by a policy maker in Zambia who said: “*[A woman] will not go to the clinic unless she is sick…So expecting healthy people to voluntarily come for vaccination is difficult*” [37]. Delayed health seeking behavior, an established trend in SSA, coupled with a lack of knowledge about the preventative nature of HPV vaccination, largely contributed to stakeholders’ understandings of cervical cancer and the HPV vaccine across studies [36,37,38].

There were significant social layers of influence on stakeholder perceptions about HPV vaccination, as HPV vaccination behaviors, actions and attitudes are shaped by both exposure to information and interaction with fellow community members [41]. Political leaders, religious figures, community leaders, community elders, health care providers, teachers, and peers all contributed to local behaviors and understandings related to HPV vaccination [24,36,37,47,48]. A stakeholder spoke to the role of political influence on perceptions of HPV vaccination saying, “*The former first lady [used to be] on television all the time talking about cervical cancer and we had an overwhelming response…The [patients] who were coming told us that [they] heard about it from the first lady*” [37]. Other prominent influential figures other than political leaders included religious leaders and community leaders, with various stakeholders expressing that communal HPV vaccination acceptance begins with such leaders [24,37,47,48]. One stakeholder also expressed the importance of teachers saying, “*…a child does not doubt [a teacher]. They can go home and convince the parent ‘this is what the teacher said*” [36]. Conversely, some adolescents said they would get the HPV vaccine regardless of layers of social influence from parents and caregivers [26,29,48].

### 3.7. Understandings of How Misinformation Is Perpetuated

Sources where people seek information on HPV vaccination is relevant to the perpetuation of misinformation. While factually inaccurate, misinformation circulating in some countries acted as both a facilitator and a barrier to the uptake of the HPV vaccine. In one focus group discussion with parents in Uganda, the notion that HPV vaccination could protect people from other viruses, such as HIV, arose amongst stakeholders; “*Since HPV and HIV are both viruses, some people believed that HPV vaccination can also prevent HIV*” [41]. Similarly, the assumption that HPV vaccination could protect some girls from getting pregnant was a reason one adolescent female cited as to why some girls get the HPV vaccine [41]. Although both of these beliefs are factually inaccurate, they facilitated the uptake of the HPV vaccine. Conversely, misinformation also acted as a deterrent to the HPV vaccine, evident in some people explaining that they could not get the HPV vaccine because “it would damage the body” or cause “future disease from the chemicals the vaccine contains”, harboring the belief that the HPV vaccine kills people slowly over time [25,41].

Misinformation characterized understandings of cervical cancer as well as HPV vaccination. Some stakeholders believed (and feared) that cervical cancer causes infertility, while others believed (and feared) that the HPV vaccine causes infertility. Misinformation was perceived to be perpetuated in communities in different ways; Nyambe and colleagues attributed persistent misinformation in Zambia to low levels of social mobilization around HPV vaccination, whereas Masika and colleagues attributed misinformation around HPV vaccination in Kenya to a lack of uniform and top-down training of relevant stakeholders ranging from policy makers, to health care providers, to teachers assisting with administering the HPV vaccine [37,40]. Insufficient comprehensive training on HPV vaccination further contributed to the gaps in rollout procedures and was considered to hinder the success of school-based HPV vaccination Programs in Zambia, Kenya, Mozambique, and Zimbabwe, where teachers (assisting health care providers administer the HPV vaccine to adolescent girls) expressed how they did not receive sufficient information or training on HPV vaccination [34,36,37,40,49]. Another female teacher from a different study in Kenya suggested: “*All teachers should be given the same information…For instance, in our district, only the headteacher and two other teachers were called*” [about HPV vaccination training] [40].

Improper training around HPV vaccination led to poor communication, which in turn, perpetuated existing misconceptions about the HPV vaccine, inevitably introducing or heightening skepticism, fear and uncertainty around HPV vaccination within communities and even in the minds of key stakeholders such as health care providers and teachers [23,34]. A lack of comprehensive information given to communities at the onset of the introduction of HPV vaccination, almost always initiated and even expedited, the spread of rumors and misinformation in its place [26,33,38,40]. Such misinformation included distortion of technical facts such as appropriate dose schedule [29,50,51], rationale for age of HPV vaccination initiation [26,49] and rumors on the preparedness and training of vaccinators [15,34].

### 3.8. Theme 2: Fear of Infertility Shaping Contradictory Perceptions of HPV Vaccination

Fear of infertility was the most salient theme in stakeholders’ narratives about cervical cancer and HPV vaccination [14,25,26,41]. Male and female stakeholders placed significant value on a woman’s ability to bear children [26,33,38,40,52], strongly fearing infertility which was highly stigmatized and seen as a source of shame and lowered female worth [26,33,38,40,52].

Two predominant narratives prevailed in relation to the etiology of infertility in relation to cervical cancer and the HPV vaccine. Some believed infertility was a result of cervical cancer [15,29,41,43] while others believed infertility was a result of receiving the HPV vaccine [26,28,29,33,34]. These contrasting perspectives contributed to differing HPV vaccination behaviors.

### 3.9. Perception That Infertility Is a Result of Cervical Cancer

Depending on the timing of presentation for care, type of HPV strain, severity of symptoms, and progression of disease, cervical cancer can cause infertility in some cases, but this is not a guarantee nor a norm [3,5]. Conversely, stakeholders in some studies believed that infertility was an inevitable outcome of cervical cancer [24,26,29,33,34,37,38,49]. A father in Kenya explained: “*When one hears the term cervical cancer especially when your child has it, you get scared. You then ask yourself whether she will ever give birth… Because when she has cervical cancer, she might not give birth and will finally die, so a parent loses hope*” [47]. Stakeholders with similar beliefs were in obvious support of HPV vaccination as a means to prevent cervical cancer and by extension, infertility [24,26,29,33,34,37,38,49].

### 3.10. Perception That Infertility Is a Result of the HPV Vaccine

Conversely, other stakeholders believed HPV vaccination was the cause of infertility [26,28,29,33,34]. Although the origin of this belief could not be identified, it was highly prevalent across studies and had significant influence on how some stakeholders’ perceived HPV vaccination [26,28,29,33,34]. Fear that the HPV vaccine causes infertility was the most prevalent justification for vaccine hesitancy, as evidenced in stakeholders expressing: “*They said it kills a woman’s eggs and she does not produce children… Some believed that the vaccine was meant to reduce fertility of women in future by destroying their ovaries…*” [41].

These two schools of thought, that cervical cancer causes infertility amongst women and conversely, that HPV vaccination causes infertility amongst women, did not seem to overlap, and both weighed significantly in stakeholder decisions around HPV vaccination in SSA [9,41,47].

The debate around the cause of infertility was further fueled by pertinent distrust of both local health systems and governments, as well as international vaccine initiatives and stakeholders from high-income countries who were perceived to be drive the roll out of HPV vaccination in some studies [13,28,33,34]. As one participant explained: “*What I have heard about the vaccine…some say that it has been developed to reduce the population, to reduce the fertility in a woman, an African woman*” [34]. Some stakeholders in Zambia explained that a lack of information given to them during the implementation of the HPV vaccination program made them question how much research had been done on the HPV vaccine, whereas some stakeholders in Uganda questioned whether HPV vaccination was an opportunity for government to conduct research and “experimentation”, using young girls as “guinea pigs” [13,37]. Others worried that HPV vaccination was just a government-supported initiative for population control of Africans or that Africa was receiving “second-tier” or “left over” vaccines from first world countries [15,33,37]. This sentiment of uncertainty was echoed by several stakeholders across studies, making an underlying suspicion of public health initiatives and widespread anti-fertility rumors a large factor when considering HPV vaccination in SSA [15,26,33,37]. Mistrust and misinformation often left stakeholders unsure of what to believe around HPV vaccination, best displayed in an adolescent female from a focus group discussion in Uganda saying: “*People say that the HPV vaccine may make us fail to bear children in future; but we have also been told that it is not true that the injection can cause infertility. So we do not know the truth*” [41].

### 3.11. Theme 3: The Feminization of HPV Vaccination in SSA

Gender dynamics, pertaining to HPV vaccination Programs, vaccination decision-making and the contexts in which these occur, also featured prominently in stakeholders’ narratives.

HPV vaccination Programs have traditionally only targeted female adolescents as women account for most morbidity and mortality from HPV-related outcomes [33]. As HPV is a gender-neutral virus, several stakeholders were confused by HPV vaccination being solely directed at females [15,23,28,36] evident in a question asked by a public health nurse in Kenya; “*Why target women and not men? Why don’t you target the source? Why don’t you put out the fire from where it starts?*” [24].

Some male stakeholders expressed disinterest in HPV vaccination, assuming that since it is a female-directed vaccine aimed at addressing cervical cancer, that HPV must not affect men [15,23,36]. The focus on cervical cancer (and by extension, the necessity of having a cervix in order to be directly affected), coupled with a traditional cultural focus of reproductive health being a part of a “woman’s job”, reinforced feminization of HPV and caused feelings of resentment, confusion, and distrust towards HPV vaccination from some stakeholders of both sexes in SSA [15,28]. Other stakeholders expressed concern that just vaccinating girls was not only discriminatory, but also confusing for boys, especially when trying to close the unequal gendered power dynamic in SSA [28,29]. Some stakeholders called for future vaccination campaigns to be gender-neutral and in the interim, with concern about the progressively early age of sexual debut of male and female adolescents, [15,28,41,48] for education campaigns to shift to a “gender-neutral focus” to emphasize the gender neutrality of HPV infection amongst current generations of adolescent males and females alike [23,43].

In some studies, female HPV vaccination was perceived to provide some protection in environments where gender-based violence and abuse of female children and adolescents was highly prevalent. For example, one female parent in South Africa said; “*I feel certain about [vaccinating my child] because there is AIDS and HIV out there and we all are aware of it… My child can be raped, and I will feel bad about it, but I am at peace [knowing] that she is participating in the Kganya Motsha [HPV vaccine trial] and will be protected against [this] sexually transmitted disease*” [48]. The inevitability of rape in South Africa was also mentioned in an interview with a Sangoma [traditional healer] [32] and a father [48]. In this manner, the HPV vaccine was seen as a harm reduction strategy against gender-based sexual violence, which some stakeholders perceived as inescapable for females in some contexts [14,32,37,43,48].

Health-seeking behavior was also shaped by gender dynamics. In some countries (Zambia, Zimbabwe, Kenya and Tanzania), men had the ultimate say in women’s health decisions, including decisions around HPV vaccination [23,26,34,36,37]. One woman from a focus group in Zambia explained this further: “*[There is] the cultural background that a woman should seek permission from her husband, whether she should take her daughter for the vaccine… so those are cultural issues that will always be there*” [34]. Some teachers in Tanzania agreed, explaining that school meetings were not always successful when it came to educating parents and caregivers about HPV vaccination for their adolescent daughters because even if the mother was convinced to get the daughter vaccinated against HPV, many fathers refused and had the final say in HPV vaccination uptake for their daughters [26]. The converse was true in South Africa and Malawi, where stakeholders agreed that the responsibility for the care of children lay solely with women, including healthcare decisions for children, while fathers and male kin were generally absent from the process [14,30].

In general, male and female stakeholders across studies urged men to be more included in education around cervical cancer and HPV vaccination. In addition to men being carriers of HPV and being susceptible to cancer outcomes such as anal, penile and throat cancer [15,24], greater male involvement could help to reduce the stigma around HPV vaccination that has been seen as a female responsibility and health issue [29]. This is further evident in the quote by one female focus group participant in Zambia saying: “…*but with education we should include the male folk because mostly we side line them, [but] they also play an important role*” [34].

## 4. Discussion

Three major themes emerged from this review. Firstly, stakeholders’ understandings and perceptions of cervical cancer, HPV and HPV vaccination are shaped by a mix of knowledge and misinformation, both of which are influenced by social processes and relationships and both of which act as both barriers and facilitators to HPV vaccination. Within the broader literature, this review supports Abdullah and colleagues that there is sub-optimal knowledge about vaccine preventable diseases and associated vaccines in Africa, albeit moderate levels of positive attitudes towards HPV vaccination [46]. The studies included in this review revealed that misinformation is fueled by both rumors and fear, which simultaneously contribute to a growing suspicion and distrust of vaccination Programs (in general and HPV vaccination specifically) in SSA. The issue of trust and distrust has received considerable attention within the broader vaccination demand literature in general, as well as in specific relation to HPV vaccination. In regard to vaccination in general, broader qualitative research suggests that trust of the vaccine, the provider, and the policy-maker are what drive vaccine acceptance and that perceived trust violations often result in various ways, one of which from the perception that health care providers, government or the wider health system financially profit from vaccination, leaving stakeholders feeling like there is a hidden agenda in the suggestion to be vaccinated [53,54,55]. This broader research finding is relevant to the findings of this review, evident when some stakeholders attributed their HPV vaccination hesitancy to skepticism of government and health care providers intentions and agenda [26,28,29,33,34]. This idea contributes to the greater findings of this review that there is an obvious social dimension to the exchange of information around HPV vaccination, with special emphasis on sources of HPV vaccination information shaping how information is perceived. This finding supports the broader literature on vaccination trust [53,54,55,56,57] and cut both ways in this review; some stakeholders placed large trust and validation in a health care provider recommendation for HPV vaccination, while others placed very little, or even diminished value, from the same health care provider recommendation. Notable is the varying level of trust of government and health care providers across stakeholders, with several stakeholders in this review perceiving HPV vaccination to be a covert method of population control and thus having a deep distrust of both the initiative of HPV vaccination and the health care providers carrying it out [26,28,29,33,34]. Conversely, some stakeholders requested validation or further information about HPV vaccination from community or religious leaders, whom were often perceived as acting in the best interest of the community (and therefore deserved trust), [24,36,37,47,48] often being suggested as figures to publicly endorse HPV vaccination programs to potentially mitigate non-acceptability barriers such as doubts about HPV vaccination efficacy and fear of safety and side effects [54]. These underlying dynamics need to be both identified and better understood. The disconnect in where and whom stakeholders choose to place their trust is notable, especially considering distrust of HPV vaccination has led to further misinformation and has left many stakeholders unable to ascertain the validity of the information about HPV vaccination in SSA.

A second central theme that emerged from this review was how beliefs around infertility have created a significant divide in understandings of HPV vaccination in SSA. Stakeholders tended to hold one of two opposing views of the causes of infertility. Some attributed infertility to cervical cancer thus encouraging support for the HPV vaccine for prevention of common cancer-causing strains of HPV and by extension, infertility. Conversely, others attributed infertility to the HPV vaccine itself, which fueled HPV vaccination hesitancy and refusal amongst stakeholders who harbor such beliefs. This supports broader findings that infertility amongst African women can have serious psychological and social implications for women, along with being a major factor for marital instability, social pressure and ostracization, stigmatization and abuse [56,57,58,59]. Examining perceptions that infertility can be caused by HPV vaccination is vital to understanding HPV vaccination hesitancy in the African context.

A final theme that emerged from this review was the significant role that gender dynamics play in how stakeholders perceive HPV vaccination. Gendered dynamics were present in stakeholder perceptions and understandings of HPV vaccination in the following ways; the notion that HPV vaccination can be a viable harm reduction strategy against female gender-based violence, significant differences across countries in SSA in regard to whom makes decisions within households about health seeking behaviors (such as HPV vaccination), and finally, that female-only HPV vaccination campaigns encourages the perception of HPV as a female specific issue which perpetuates misinformation. This is aligned with the broader research examining the feminization of HPV due to the focus on cervical cancer screening, which has not only made females the primary group responsible for HPV prevention, but has also resulted insufficient protection from HPV-related illnesses in males and females alike [55,56,57,60]. This review also supports the broader literature of gendered disparities in HPV vaccination uptake by sex being consistent with the delayed recommendation for boys to also get the HPV vaccine, and today, further evident in a lack of coverage for boys in many National Insurance Programs with conflicting messages around gender norms as both a barrier and facilitator to HPV vaccination uptake [55,56,57,60,61].

The findings from this review have various practical implications for how demand for HPV vaccination might be enhanced. The dynamics of trust and drivers of distrust between communal stakeholders and the government, medical and public health community need to first be recognized and then be addressed through wide spread meaningful engagement. Prior to rollout, education and social mobilization around the HPV vaccination Programs should sufficiently engage communities in order to thoroughly gauge local understandings, conceptions and misconceptions, language nuances, baseline perceptions and fears around cervical cancer, HPV and HPV vaccination. Utilizing local language nuances of disease terminology and recognized symptomology may be an important consideration for HPV vaccination roll out strategies across SSA when shaping content and focus of education and awareness-raising interventions [26,31,32,33,37,43]. Secondly, engagement with and involvement of various social actors identified as trusted sources of information from the community themselves, may facilitate improvements in stakeholder perceptions of HPV vaccination in SSA. This is especially notable in the effort to mitigating the belief that HPV vaccination causes infertility, the most prominent and wide-spread rumor across SSA. Finally, implementation of HPV vaccination Programs without thorough understanding of gender dynamics within the context of the specific country implementing HPV vaccination can potentially create roadblocks for HPV vaccination uptake, as well as perpetuate confusion, misinformation, distrust and stigma. This is notable as this review found that, depending on the country, health seeking behavior and decision making around HPV vaccination were driven by either male or female head of households, suggesting that locally appropriate interventions need to speak to country specific gender dynamics in health seeking behavior.

Although some studies used focus group discussions and interviews to assess preliminary perceptions of cervical cancer, HPV, and HPV vaccination, often following up with an educational component at the completion of the study to address misinformation, it is evident that efforts to address misinformation require first understanding the social context and the social dynamics within that context, as identified in this review. This supports previous research that community acceptability towards HPV vaccination is a slow process that requires first acknowledging rumors, fears, and concerns, prior to expressing the benefits and safety of HPV vaccination [41]. This review supports prior research that calls for strategies to first address prominent misinformation about HPV vaccination in the SSA context prior to HPV vaccination campaign roll out [13,14,16,26,41], highlighting that HPV vaccination decision-making is complex and multifactorial [60,61].

### 4.1. Reflexivity

The lead researcher has a background in public health and acknowledges that this has shaped positive attitudes towards vaccination more generally. As cervical cancer is largely preventable, the lead researcher undertook this analysis in order to better understand extreme rates of morbidity and mortality from HPV and cervical cancer in SSA [21].

### 4.2. Strengths and Limitations of the Rapid Qualitative Review

A strength of this review is the consideration and acknowledgement of divergent findings within the dataset and what such conflicts reveal about understandings of HPV vaccination in SSA. This review provides a ‘big picture’ demand-side perspective to contribute to improvements in current and future HPV vaccination strategies in SSA, an under-researched area overall.

This was a rapid qualitative review and thus has various limitations common to this approach. Limitations include: a single review author for all stages of the search strategy (although this limitation was addressed through a 20% random sampling of title and abstracts and eligible full text studies, assessed by two independent reviewers). A ‘Confidence in Evidence from Reviews of Qualitative Research’ assessment was not conducted [62]. The date of studies ranges from 2006–2018, introducing the possibility that some of the previously published literature is out of date compared to present-day. Exclusion of grey literature is a limitation due to time and resource constraints, as well as exclusion of non-English studies due to the language capacities of the review team.

## 5. Conclusions

Stakeholders who create a demand for the HPV vaccine are arguably the most vital to its uptake and continued necessity, especially when the momentum of HPV vaccination in SSA has only recently been initiated. How stakeholders understand cervical cancer, HPV, and HPV vaccination will be vital to the short- and long-term success of HPV vaccination Programs in SSA. This review found that stakeholder understandings of HPV vaccination are shaped by a complex relationship between knowledge and misinformation, a significant fear of infertility associated with both cervical cancer and the HPV vaccine, and social and gender dynamics in SSA. This review iterates the importance of first working with communities to gauge local and context-specific understandings, before trying to implement change through one-size-fits all education, sensitization and behavior change strategies.

## Figures and Tables

**Figure 1 vaccines-09-00496-f001:**
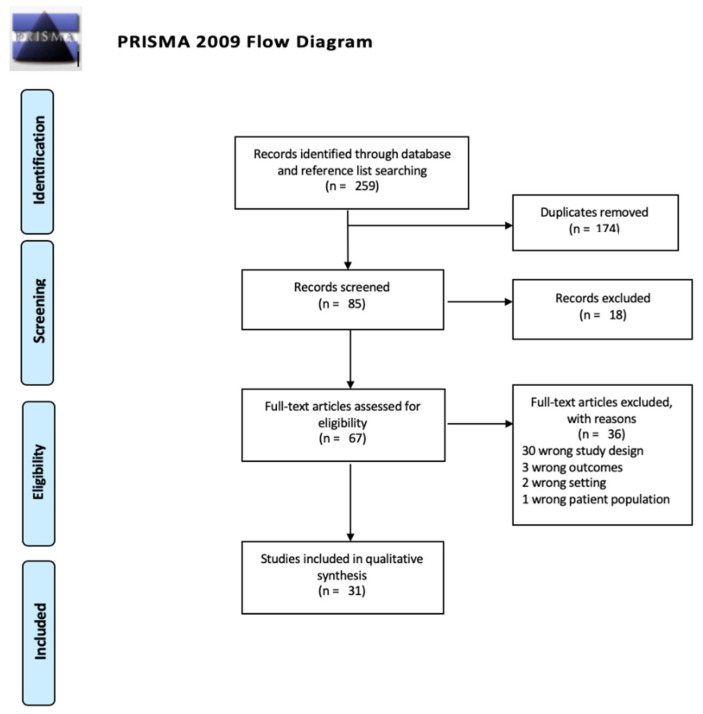
PRISMA diagram of search strategy and results [20].

**Figure 2 vaccines-09-00496-f002:**
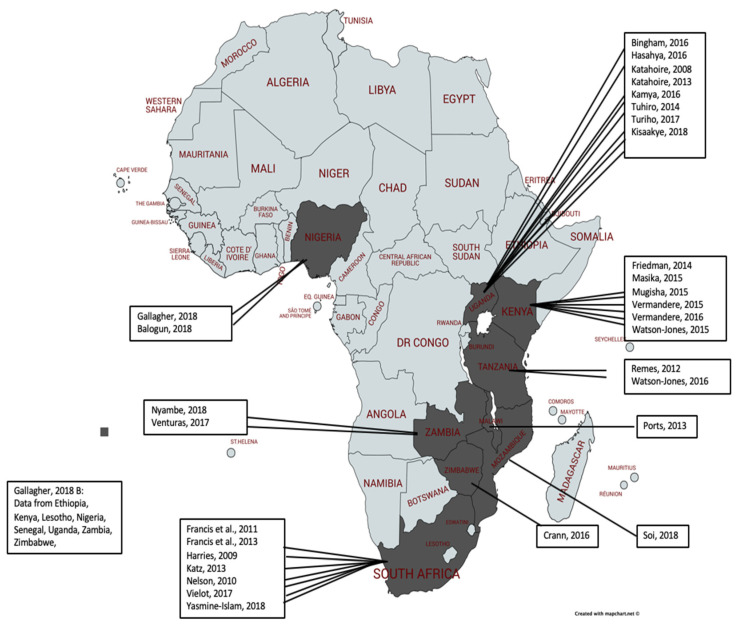
Included studies in thematic analysis mapped by country of origin (first author, year).

**Table 1 vaccines-09-00496-t001:** Summary of themes and findings.

Theme	Main Points:
Theme 1: Knowledge and misinformation	- Generally low levels of technical knowledge about cervical cancer and HPV in SSA -Knowledge and misinformation about cervical cancer and the HPV vaccine exist in parallel and cause confusion in stakeholder perceptions of HPV vaccination -Cervical cancer is widely feared and stakeholders expressed clear desires to avoid a cerivcal cancer diagnosis - There were significant social layers of influence on stakeholder perceptions about HPV vaccination, as well as actions and attitudes that impact HPV vaccination behaviour - Misinformation circulating in some countries acted as both a facilitator and a barrier to the uptake of the HPV vaccine
Theme 2: Fear of infertility shaping contradictory perceptions of HPV	- Fear of infertility was the most salient theme in stakeholders’ narratives about cervical cancer and HPV vaccination - Two predominant narratives prevailed in relation to the etiology of infertility in relation to cervical cancer and the HPV vaccine. Some believed infertility was a result of cervical cancer, while others believed infertility was a result of receiving the HPV vaccine
Theme 3: The feminization of HPV vaccination in SSA	- Gender dynamics featured in stakeholder narratives about HPV vaccination Programs and HV vaccination decision-making. - HPV is a gender-neutral virus, yet several stakeholders were confused why HPV vaccination is solely directed at females in the SSA context -Health seeking behaviour around HPV vaccination is often influenced by gender dynamics -In general, male and female stakeholders across studies urged for males to be included in education around cervical cancer and HPV vaccination

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
