# Peer review of "Stakeholders’ Understandings of Human Papillomavirus (HPV) Vaccination in Sub-Saharan Africa: A Rapid Qualitative Systematic Review"

_vaccines, 2021, doi:10.3390/vaccines9050496_

Round 1

Reviewer 1 Report

Article is fine, would only check again for any remaining mild spelling and grammar mistakes.  

Author Response

Reviewer One: Article is fine, would only check again for any remaining mild spelling and grammar mistakes

Author Response to Reviewer one: I thank the reviewer for this feedback and have checked for any spelling and grammar mistakes- Any spelling or grammar changes made are evident in tracked changes in the uploaded version of the manuscript.

Reviewer 2 Report

The Systematic Review manuscript entitled “Stakeholders’ Understandings of Human Papillomavirus (HPV) vaccination in Sub-Saharan Africa: A Rapid Qualitative Systematic Review” by Deignan and colleagues provide an overview on the stakeholders’ understandings and perceptions of cervical cancer, HPV and HPV vaccination in general in Sub-Saharan Africa.

  1. The analysis is potentially interesting and will improve our knowledge on the HPV vaccination programs of the Sub-Saharan Africa. However, the work should be shortened for at least 30%. Several part should be shortened, for instance both “Reflexivity as well as Strengths and limitations” sections are quite long. The method section, especially “Data extraction and analysis” is very long. To reduce redundancy and length may I suggest removing the patients/females quotes within the text?
  2. General question. What about HPV-driven vulvar cancer (PMID: 32321580). Has this tumor been mentioned in the analyzed works?

Minor comments

Line 11-14 The vaccine type should be detailed (bivalent, quadrivalent or nonavalent?). Lines 42-55. This information should included in this section as well. “The HPV vaccine was first introduced in 2006”--> the specific vaccine type and where (Africa? Worldwide?) should be both detailed. For instance, has the nonavalent been introduced in 2006? These are helpful papers in this field PMID: 32154146, PMID: 29266694, PMID: 25875167, MID: 24812407, PMID: 24229716 PMID: 33571186 

Line 37 these references should be included (PMID: 30650383; PMID: 32854278; PMID: 33339760; PMID: 28357382)

Line 90 Regarding the aim It should be more professional a sentence without questions

Line 97-100 this sentence is quite repetitive as almost identical to the above sentence

Line 126-138 The Study Selection can be shortened as Fig already describe the paper selection process. The points are almost obvious, especially point 1. They can be shortened by directly quoting the PRISMA guidelines. In addition, author initials should be removed

Author Response

Reviewer two:

  1. The analysis is potentially interesting and will improve our knowledge on the HPV vaccination programs of the Sub-Saharan Africa. However, the work should be shortened for at least 30%. Several part should be shortened, for instance both “Reflexivity as well as Strengths and limitations” sections are quite long. The method section, especially “Data extraction and analysis” is very long. To reduce redundancy and length may I suggest removing the patients/females quotes within the text?

Author Response: Thank you for this feedback. I have removed unnecessary detail evident in tracked changes on the manuscript:

  • Reflexivity section: Removed several lines (evident in tracked changes) before Line 597
  • Strengths and limitations section: Removed several lines (evident in tracked changes) before Line 615
  • Data Extraction and analysis section: Significantly reduced this section (evident in tracked changes) before line 175

Given that Vaccines has no restrictions on the length of manuscripts, provided that the text is concise and comprehensive, I have removed quotes where they are redundant (evident in tracked changes in line 329, line 345, line 383, line 400, line 417, line 461, & line 508). Otherwise, I feel that the included quotes are needed to highlight and legitimize the findings in the qualitative analysis. I hope the reviewer finds the reductions in the reflexivity section, strengths and limitations section, and data extraction and analysis section sufficient in order to shorten the length of the manuscript.

  1. General question. What about HPV-driven vulvar cancer (PMID: 32321580). Has this tumor been mentioned in the analyzed works?

Author Response: Thank you for this feedback. HPV-driven vulvar Cancer has explicitly now been mentioned in line 37

Minor comments:

  • Line 11-14 The vaccine type should be detailed (bivalent, quadrivalent or nonavalent?).
    • Line 11-14 has been amended to include the type of vaccine and additional information on the protection against type 16 and 18 of HPV known to cause 70% of cervical cancers
  • Lines 42-55. This information should included in this section as well. “The HPV vaccine was first introduced in 2006”--> the specific vaccine type and where (Africa? Worldwide?) should be both detailed. For instance, has the nonavalent been introduced in 2006?
    • These are helpful papers in this field PMID: 32154146, PMID: 29266694, PMID: 25875167, MID: 24812407, PMID: 24229716 PMID: 33571186
    • Numbering of lines has changed to 55-60 and vaccine type has been detailed along with additional geographical information about vaccine usage, evident in tracked changes.
  • Line 37 these references should be included (PMID: 30650383; PMID: 32854278; PMID: 33339760; PMID: 28357382)
    • PMID: 33339760 was added as a Reference
  • Line 90 Regarding the aim It should be more professional a sentence without questions
    • Now Line 123:The sentence has been re-written to be a clear, professional statement, evident in tracked changes of line 123
  • Line 97-100 this sentence is quite repetitive as almost identical to the above sentence
    • Now lines 123-132: Deleted lines 123-127 to reduce redundancy and make it one coherent thought/sentence about justification for the review we conducted, evident in lines 123-128.
  • Line 126-138 The Study Selection can be shortened as Fig already describe the paper selection process. The points are almost obvious, especially point 1. They can be shortened by directly quoting the PRISMA guidelines. In addition, author initials should be removed
    • Now Lines 154-161: The study selection section has been shortened and redundancies of point 1-4 have been removed, creating a coherent sentence that refers to the PRISMA figure for study selection process.

Reviewer 3 Report

This manuscript described stakeholders’ understanding of HPV vaccination in Sub-Saharan Africa. It is well organized; however, it is better to add several points as shown below.

 The authors mentioned the knowledge and perception of cervical cancer and HPV vaccine, fear of infertility by HPV vaccination, and the feminization of HPV vaccination. It is better to summarize the main points by theme in tables for a better understanding of this manuscript. It is also better to conduct a numerical analysis if possible.

Author Response

Reviewer three: This manuscript described stakeholders’ understanding of HPV vaccination in Sub-Saharan Africa. It is well organized; however, it is better to add several points as shown below.

The authors mentioned the knowledge and perception of cervical cancer and HPV vaccine, fear of infertility by HPV vaccination, and the feminization of HPV vaccination. It is better to summarize the main points by theme in tables for a better understanding of this manuscript. It is also better to conduct a numerical analysis if possible.

Author response to reviewer three:

Thank you for this feedback. Line 347-348: Per the reviewer’s request, we included a brief table of main points by theme. In regard to the numerical analysis suggestion, as this was a qualitative evidence synthesis that adhered to the methods for conducting a qualitative systematic review, a numerical analysis within the review is not possible.

Round 2

Reviewer 2 Report

The authors have addressed all my concerns and therefore I support publication

Reviewer 3 Report

The authors revised the manuscript according to the reviewer's comment. It is considered suitable for publication.